# Assessing SARS-CoV-2 Vertical Transmission and Neonatal Complications

**DOI:** 10.3390/jcm10225253

**Published:** 2021-11-11

**Authors:** Cosmin Citu, Radu Neamtu, Virgiliu-Bogdan Sorop, Delia Ioana Horhat, Florin Gorun, Emanuela Tudorache, Oana Maria Gorun, Aris Boarta, Ioana Tuta-Sas, Ioana Mihaela Citu

**Affiliations:** 1Department of Obstetrics and Gynecology, “Victor Babes” University of Medicine and Pharmacy Timisoara, 2 Eftimie Murgu Square, 300041 Timisoara, Romania; citu.ioan@umft.ro (C.C.); bogdan.sorop@gmail.com (V.-B.S.); gorun.florin@umft.ro (F.G.); 2ENT Department, “Victor Babes” University of Medicine and Pharmacy Timisoara, 2 Eftimie Murgu Square, 300041 Timisoara, Romania; horhat.ioana@umft.ro; 3Department of Pulmonology, “Victor Babes” University of Medicine and Pharmacy Timisoara, 2 Eftimie Murgu Square, 300041 Timisoara, Romania; tudorache_emanuela@yahoo.com; 4Department of Obstetrics and Gynecology, Municipal Emergency Clinical Hospital Timisoara, 1-3 Alexandru Odobescu Street, 300202 Timisoara, Romania; oanabalan@hotmail.com; 5Department of Obstetrics and Gynecology, Timisoara County Emergency Clinical Hospital, 12 Victor Babes Street, 300226 Timisoara, Romania; aris.boarta@gmail.com; 6Discipline of Hygiene, Department 14 Microbiology, “Victor Babes” University of Medicine and Pharmacy Timisoara, 2 Eftimie Murgu Square, 300041 Timisoara, Romania; tuta-sas.ioana@umft.ro; 7Department of Internal Medicine I, “Victor Babes” University of Medicine and Pharmacy Timisoara, 2 Eftimie Murgu Square, 300041 Timisoara, Romania; citu.ioana@umft.ro

**Keywords:** SARS-CoV-2, COVID-19, pregnancy, intrauterine infections, vertical transmission

## Abstract

We designed and implemented a prospective study to analyze the maternal and neonatal outcomes associated with COVID-19 and determine the likelihood of viral transmission to the fetus and newborn by collecting samples from amniotic fluid, placenta, umbilical cord blood, and breast milk. The study followed a prospective observational design, starting in July 2020 and lasting for one year. A total of 889 pregnant women were routinely tested for SARS-CoV-2 infection in an outpatient setting at our clinic, using nasal swabs for PCR testing. A total of 76 women were diagnosed with COVID-19. The positive patients who accepted study enrollment were systematically analyzed by collecting weekly nasal, urine, fecal, and serum samples, including amniotic fluid, placenta, umbilical cord, and breast milk at hospital admission and postpartum. Mothers with COVID-19 were at a significantly higher risk of developing gestational hypertension and giving birth prematurely by c-section than the general pregnant population. Moreover, their mortality rates were substantially higher. Their newborns did not have negative outcomes, except for prematurity, and an insignificant number of newborns were infected with SARS-CoV-2 (5.4%). No amniotic fluid samples were positive for SARS-CoV-2, and only 1.01% of PCR tests from breast milk were confirmed positive. Based on these results, we support the idea that SARS-CoV-2 positive pregnant women do not expose their infants to an additional risk of infection via breastfeeding, close contact, or in-utero. Consequently, we do not support maternal–newborn separation at delivery since they do not seem to be at an increased risk of SARS-CoV-2 infection.

## 1. Introduction

Severe Acute Respiratory Syndrome Coronavirus 2 (SARS-Cov-2) was initially identified in December 2019 in the Wuhan region of China and quickly spread to the rest of the world and, within a few weeks, being declared a pandemic. Romania reported its first Coronavirus Disease 2019 (COVID-19) cases in February 2020, and about 1.5 million cases have been documented as of October 2021 [1]. COVID-19 is a respiratory illness caused by the SARS-CoV-2 virus that is more severe than the seasonal flu [2], causing around 5% of the patients diagnosed with this pathology to require ICU hospitalization and about 3% to succumb [3]. In these patients, the “hyperchromatic supranuclear stria” (SNS), corresponding to the Golgi apparatus, constitutes a marker for the anatomical and functional integrity of the ciliated cells; its rarefaction or disappearance during viral infections is a sign of cellular distress [4]. COVID-19 infection has an insidious course, often behaving in children or adolescents in a paucisymptomatic or asymptomatic manner, increasing the risk of contagion in the general population [5].

To the present day, the available data reveal no significant differences in the clinical symptoms of COVID-19 infection between pregnant and non-pregnant women or adults of reproductive age [6,7]. On the other hand, pregnant women may be at an increased risk of developing severe illnesses that necessitate admission to a maternal intensive care unit and mechanical ventilation as a result of respiratory infection, compared to the general population [8]. Pregnancy does not seem to enhance susceptibility to SARS-CoV-2 infection, and most infected pregnant patients recover before giving birth [9], although pregnant women with COVID-19, particularly those who develop pneumonia, tend to have a higher rate of premature birth and cesarean delivery [10,11]. Evidence on several aspects of the prenatal management of these pregnancies remains mixed, including the type and frequency of fetal monitoring, the potential risk associated with invasive prenatal diagnosis, the timing of delivery, and intrapartum monitoring [12].

Although some viruses that target the human respiratory tract, such as the syncytial respiratory virus, are known for their potential vertical transmission [13], there is no research indicating significant risks of SARS-CoV-2 being vertically transferred to the fetus [14]. The placenta physiologically limits vertical transmission during pregnancy and has developed robust microbial defense mechanisms. It remains, to this day, uncertain whether the SARS-CoV-2, among the various microorganisms that cause congenital diseases, may have evolved different mechanisms to circumvent these defenses [15]. The risk of vertical transmission from infected mothers to their neonates was assessed during previous coronavirus outbreaks, but no significant number of cases of vertical transmission has been reported to date [16]. Previous research that examined the vertical transmission of SARS-CoV-2 indicates no significant evidence of vertical transmission, although it was feasible and could not be ruled out because the virus was detected in amniotic fluid and breast milk [17], despite the danger seeming to be modest regarding fetal effects [18,19]. The evidence on SARS-CoV-2’s impact on pregnant women reveals the percentages of mild, moderate, and severe disease to be comparable to those in the general population [20]. In contrast, others suggest that pregnant women over 20 weeks’ gestation have higher Intensive Care Unit (ICU) admission rates and need oxygen supplementation when diagnosed with COVID-19 [21]. Another consequence of COVID-19 in pregnancy seems to be a preeclampsia-like condition that has been documented in the second and third trimesters [22]. However, research regarding the impact of SARS-CoV-2 infection during the first trimester of pregnancy is scarce, with only a few papers indicating low rates of miscarriage during the first trimester [23].

A World Health Organization’s (WHO) scientific brief considered three modes of vertical transmission: (1) in utero, where the virus is present in the blood and crosses the maternal-placental interface; (2) intrapartum, which occurs during labor and childbirth via contact with maternal blood, vaginal secretions, or feces; and (3) postnatal, which occurs via breast milk but can also include contact with another infected mother [24]. The WHO evaluation emphasized the difficulty of identifying vertical transmission of SARS-CoV-2 since most of the reported vertical transmission cases have been predicated on a single positive neonatal RT-PCR in an upper respiratory tract specimen, with substantial heterogeneity in the sample collection time. A proposed definition of confirmed in utero transmission is a postpartum positive RT-PCR test in one or more neonatal samples, including neonatal blood, nasal aspirate, urine or stool samples, born from mothers who tested positive for SARS-CoV-2 during the third trimester [25]. On the other hand, vertical transmission was defined by the UK Obstetric Surveillance System (UKOSS) as a positive neonatal sample collected during the first 12 h after delivery to a woman with a proven SARS-CoV-2 infection [26].

As the pandemic is evolving, obstetricians face significant challenges in treating and managing pregnant women who are affected by SARS-CoV-2, since guidelines are constantly changing or are unavailable, and the evidence available to date remains limited. Therefore, we conducted prospective observational research at a referral center for pregnant women infected with COVID-19 during the pandemic to comprehensively evaluate the evidence suggesting vertical transmission of SARS-CoV-2. We also planned to examine the impact of COVID-19 on pregnancy and the influence of pregnancy status on the progression of the SARS-CoV-2 illness.

## 2. Materials and Methods

The research followed a prospective observational design conducted among pregnant women who had to suffer a SARS-CoV-2 infection during pregnancy and were admitted or investigated at the Obstetrics and Gynecology Clinic of the Timisoara Municipal Emergency Hospital between 1 July 2020 and 1 July 2021. The project consisted of building the cohort and data collection, including the neonatal follow-up, covering a minimum of 4 to 6 weeks postpartum for breastfeeding and breast milk evaluation, to establish the rate of neonatal vertical transmission through breast milk. Given the difficulties associated with defining vertical transmission and the fact that many studies and guidelines do not report the criteria used to define it, vertical transmission was defined in the current research as a positive COVID-19 test in the neonate. This includes the RT-PCR testing of respiratory tract samples, urine, feces, or serum, at birth or within the first 24 h of life, in which the mother tested positive for SARS-CoV-2 or had a documented diagnosis of COVID-19. Vertical transmission through breastfeeding was considered as a positive RT-PCR test in breastfed neonates born from mothers positive with SARS-CoV-2 in their breast milk samples.

The study was approved by the Ethics Committee of the “Victor Babes” University of Medicine and Pharmacy (Timisoara, Romania, approval no. 6664/15 June 2020) and by the Ethics Committee of the Timisoara Municipal Hospital (approval no. I-15505/15 June 2020). The Obstetrics and Gynecology Clinic of the Timisoara Municipal Emergency Hospital is a university medical unit located in Western Romania, being the largest in this region. According to the national census, this region has more than 400 thousand women of reproductive age. Considering this population size and an average COVID-19 incidence of 7% in the general population residing in Western Romania at the time of the study, we calculated the optimal sample size to an estimate of 101 pregnant women infected with SARS-CoV-2, for a confidence coefficient of 95%. Our researchers recruited 154 eligible patients during the study period; however, 78 were lost for consistent investigations, as required by our study protocol, or they reconsidered their decision of participating in this research, leaving a cohort of 76 patients, which was lower than the estimated sample size, but still sufficient. The study’s exclusion criteria include unwillingness to participate, underage pregnant women, and trouble comprehending informed consent.

Pregnant women presenting to our clinic with typical COVID-19 symptoms such as fever, cough, dyspnea, loss of smell and taste, fatigue, myalgias, headaches, and digestive symptoms, or those in close contact with a confirmed COVID-19 case, were tested for SARS-CoV-2 using an RT-PCR from nasopharyngeal and oropharyngeal smears. All pregnant women having a positive RT-PCR for SARS-CoV-2 throughout their pregnancy or two weeks before conception and neonates born to infected mothers were considered eligible for inclusion, with the purpose of being observed for a further four weeks after birth. To determine whether or not there was a danger of vertical transmission, samples of feces, urine, serum, amniotic fluid, umbilical cord blood, peripheral blood, placenta, and breast milk were analyzed according to WHO guidelines and other safety protocols [24,27]. All included patients were monitored weekly in an outpatient setting until birth, and for four weeks postpartum. The births from SARS-CoV-2 positive mothers followed the Romanian COVID-19 pregnancy and delivery guidelines [28], updated in October 2020. The same official recommendations suggest that neonates should be taken from their mothers, without benefitting from skin-to-skin contact, and held in isolation until both the mother and the newborn are confirmed with a negative RT-PCR test. However, pregnant women have the right to decide over these recommendations; in this case, they were educated on protecting their newborns from contracting the disease. All mothers who enlisted and continued to follow this study kept permanent contact with their newborns and were breastfeeding.

The parameters considered for assessment comprised: COVID-19 symptoms, preeclampsia, preterm delivery, hospital admission, ICU admission, maternal mortality, frequency of fetal morbidity and mortality, SARS-CoV-2 viral assessment in blood cord, placenta, amniotic fluid, urine, feces, neonatal morbidity, and mortality. Neonatal morbidity was considered for neonatal SARS-CoV-2 infection, pneumonia, sepsis, and ICU admission. Data collection for analyzing the previously specified variables was made possible by evaluating pregnant women for SARS-CoV-2 from nasopharyngeal and oropharyngeal swabs using an RT-PCR test until they returned a negative result. The RT-PCR test was performed every seven days from the first contact with a symptomatic pregnant woman, including testing at delivery and postpartum. The RT-PCR test was also performed prepartum from biological fluids, including the peripheral blood, amniotic fluid, urine, and feces. Postpartum, the RT-PCR test for SARS-CoV-2 was sampled from amniotic fluid, placenta, umbilical cord, and breast milk, including the follow-up period of four weeks postpartum, after a confirmed negative result. The newborns were tested using samples from nasopharyngeal aspirate, urine, and feces, and they were also followed for four weeks. Maternal general characteristics and outcomes were compared to a cohort of pregnant women who were not infected with SARS-CoV-2 during their pregnancy and were registered to our clinic during the same study period. Gestational age was calculated using the first day of the last menstrual period or using a first-trimester dating ultrasound scan. The following sonographic parameters were used to estimate fetal weight and size according to ISUOG guidelines [29]: biparietal diameter (BPD), head circumference (HC), abdominal circumference (AC), and femur diaphysis length (FDL). The fetus was considered small for their gestational age (SGA) when their weight was equal to or less than the 10th percentile. Preterm birth was defined as newborn birth at less than 37 weeks of gestation.

Statistical analysis was performed using the IBM SPSS v.26 statistical software. We calculated the absolute and relative frequencies, the mean and median values, and the associated standard deviation (SD) or interquartile range (IQR). The χ^2^ test and Fisher’s exact test were performed to compare proportions of the study variables, while Student’s *t*-test and the Mann–Whitney U-test were used to compare group differences in parametric and nonparametric data, respectively.

## 3. Results

The general characteristics of pregnant women compared in this study did not differ between the two groups, and they did not significantly differ by pre-gestational comorbidities. However, at the level of maternal outcomes, we observed significantly more cases of gestational hypertension in COVID-19 mothers (9.2% vs. 4.1%, *p*-value = 0.038), and more emergency c-sections (23.6% vs. 14.9%, *p*-value = 0.043). Four pregnant women (5.2%) from the COVID-19 group died during the study, versus 14 (1.7%) in the control group (*p*-value = 0.036). Of the four maternal deaths, two occurred during pregnancy and were COVID-19 positive, one occurred peripartum, and the other one occurred postpartum. The two maternal deaths from the COVID-19 group had severe preeclampsia. Both of them presented worsening respiratory failure antenatally and had a respiratory failure that required mechanical ventilation, but later succumbed. The third woman died peripartum during an emergent c-section, despite intensive supportive care. The fourth woman who died in the COVID-19 group developed severe respiratory complications within 7 days of an uneventful delivery and died shortly after.

None of the three neonatal deaths observed from mothers with COVID-19 were infected with SARS-CoV-2. The cause of death was attributed to prematurity complications. A total of 74 live births were registered and further followed for PCR testing four weeks after birth. Comparing this cohort of newborns with those from the control group, we did not observe many significant differences, except for prematurity (born at <37 weeks of gestation), which affected 18.9% of newborns, compared to the normal 8.8% (*p*-value = 0.006), and growth restriction (9.2% in the COVID-19 group vs. 3.9% in the control group; *p*-value = 0.031), (Table 1).

The rate of positive PCR tests among pregnant women investigated by our clinic was 8.54% (76 out of 889 patients). A total of 889 PCR tests were performed from nasal swabs. In contrast, the other PCR tests from urine, feces, serum, placenta, umbilical cord, amniotic fluid, and breast milk were conducted only for the mothers who were confirmed positive at the initial nasal swab PCR test. These tests were done on a weekly basis until found negative for SARS-CoV-2. The average onset of COVID-19 symptoms in pregnant women was at 28 weeks of gestation (SD = 7.07 weeks), with an average date of diagnosis from nasal swab PCR at 30 weeks of gestation (SD = 5.43 weeks). The IgM serology was determined positive for SARS-CoV-2, on average, at around 31 weeks of gestation (SD = 5.03 weeks) or one week after the nasal swab positivity. The admission at the inpatient clinic for follow-up was not mandatory, and it was necessary, on average, at 35 weeks of gestation (SD = 3.22 weeks) for amniocentesis and expectancy until labor onset or emergent c-section. The onset of labor was reported at around 38 weeks of gestation (SD = 3.61 weeks), and it was naturally occurring or induced due to maternal complications of fetal tracing abnormalities. Breast milk samples were found positive for SARS-CoV-2 at around 39–40 weeks of gestation or one week after labor (SD = 3.27 weeks). Hospital discharge happened on average 4 to 6 weeks after labor onset. A detailed timeline for data collection, including PCR testing and follow-up, is described in Figure 1. Finally, the pregnant women confirmed with SARS-CoV-2 died at a median of 36 weeks of gestation (IQR = 12.5 weeks). The other results from maternal PCR tests are further included and described in Figure 2.

From the four newborns confirmed with SARS-CoV-2, symptom onset was at five days postpartum (SD = 1.14 days), with the PCR test from nasal aspirate and serum showing positive results at around 1 or 2 days from symptoms onset. The other results from newborn PCR tests are further included and described in Figure 3.

From the 76 symptomatic and asymptomatic pregnant women confirmed with SARS-CoV-2 infection, 72 (97.7%) were discharged healthy. Our analysis determined that the viral clearance was represented by a median of 13 days (IQR = 9–16). From the 74 live births, only four newborns were infected with SARS-CoV-2 (5.4%), and their viral clearance was insignificantly shorter than their mothers (median = 11 days; IQR = 8–14; *p*-value = 0.649).

Nasal swab test samples were collected and analyzed from 889 pregnant women. They were positive in 76 cases (8.54%), and systematic surveillance was implemented on a weekly basis with sample collections from urine, feces, and serum. Amniotic fluid, placenta, umbilical cord, and breast milk were evaluated using PCR tests at hospital admission and postpartum. The results are presented in Figure 2, and they show that the placenta was the most common to be found positive for SARS-CoV-2 in 7.53% of the tests. Fecal samples were positive in 6.18% of the tests evaluated, followed by SARS-CoV-2 IgM serum tests with a positivity rate of 4.38%. The urine samples were only 0.89% positive. Surprisingly, the amniotic fluid analysis showed no cases of infection in all the tests performed. However, the umbilical cord samples were positive in 2.92% of cases. Eventually, breast milk was found to be a safe feeding source, with just 1.01% of positive PCR test results for SARS-CoV-2.

Nasal, urine, fecal, and serum samples were collected and analyzed from all 74 newborns from positive COVID-19 mothers. The nasal swabs were positive in 5.40% of the tests, followed by fecal (4.05%), serum (1.35%), and lastly, urine being negative in all instances (Figure 3).

## 4. Discussion

Due to the unique nature of the maternal-fetal relationship, illness and death may impact both. This prospective longitudinal research allowed the collection of consecutive instances of COVID-19 in pregnant women from Romania, achieving the goal of determining if pregnancy affects the illness’ prognosis and, conversely, whether the infection affects the pregnancy outcomes and the newborn by providing details about complications, malformations and, most importantly, the risks of vertical transmission. To our knowledge, the current study is among the first to systematically analyze the data mentioned above for observing if SARS-CoV-2 is a potential threat to the fetus and newborn by vertical transmission.

The primary merit of this research is the weekly sequential collection of nasopharyngeal and oropharyngeal smears, which allowed for the inference of viral clearance. Collecting additional samples from peripheral and umbilical cord blood, serum, feces, urine, amniotic fluid, placenta, and breast milk, as well as the serologic follow-up, provides more information about the virus’s behavior in other biological fluids and the risk of perinatal transmission than other available data. We found no evidence of vertical transmission in late pregnancy. The potential for vertical transmission of COVID-19 infection in pregnant women is relevant for understanding the clinical features and the significance and possible implications of the pathogen [30].

At present, there is limited information on the potential transmission of the infection from mother to child, particularly through breast milk and breastfeeding. A recent systematic review included 340 records, 37 with breast milk samples and 303 without [31]. The 37 articles with breast milk samples analyzed reported 77 mothers nursing their infants. Among them, 19 of 77 infants were confirmed COVID-19 cases based on RT-PCR tests, including 14 infants and five older children. Nine of 68 breast milk samples tested from mothers with COVID-19 were positive for SARS-CoV-2 RNA; of the exposed infants, four were positive and two were negative for COVID-19. Currently, there is no evidence of SARS-CoV-2 transmission through breast milk. The authors concluded that studies with longer follow-up periods that collect data on infant feeding practices and viral presence in breast milk are needed. Another review evaluated the same concern as debated by our study [32]. A total of 336 newborns were evaluated for COVID-19 and the RT-PCR positivity rate from throat swabs was about 4%. These results are consistent with our findings. An interesting observation was that all infants with positive tests for COVID-19 were delivered by c-section. Among newborns with a throat swab returning positive for SARS-CoV-2, only five had concomitant placenta, amniotic fluid, and cord blood samples examined, of which only one amniotic fluid sample was positive at RT-PCR analysis. Five neonates showed increased IgG and IgM but did not have their intrauterine tissue examined. Four newborns had chest imaging indicative of COVID-19 pneumonia. Compared to what was reported in this review, one difference is that we did not perform chest imaging on pregnant women or their newborns due to safety concerns and refusal.

More recent data were collected by Puddemann et al. [33] to assess the hypothesis of SARS-CoV-2 vertical transmission. Their results indicate that vertical transmission is feasible but uncommon, and the mechanisms that govern whether vertical transmission occurs are unclear. Furthermore, there seems to be no evidence linking maternal symptoms to vertical transmission. The mode of delivery does not correlate with vertical transmission rates. Seven of the twenty-five investigations found SARS-CoV-2 in placental tissue; however, many of these studies failed to confirm vertical transmission to the newborn. Similarly, a very comprehensive review from Brazil, a country affected by a constantly high rate of COVID-19 cases, reported that COVID-19 exposure during pregnancy may result in bad outcomes for the mother, fetus, and newborn. Additionally, exposure of neonates to the SARS-CoV-2 virus during pregnancy and delivery cannot be excluded. Still, it was only reported in 13 cases from placenta PCR tests, six positive tests from breast milk, and just 2.7% of newborns infected with SARS-CoV-2 [34]. Finally, other data supporting our findings of low SARS-CoV-2 transmission from the mother to the newborn were presented by a meta-analysis, although it was published one year prior to our study [35]. Their data show that nasal swabs from newborns were positive in only 3.2% of instances, and the amniotic fluid samples were all negative, like in our study. The researchers affirm that the infection rates are comparable to those of other pathogens known to cause congenital illnesses.

One limitation faced by our study is the relatively small sample size. Although many pregnant women were evaluated, the rate of SARS-CoV-2 infection at the time of the study did not increase drastically, thus having an 8–9% incidence among pregnant women. Moreover, among these positive cases, we managed to include and systematically follow just 76 pregnant women. Another limitation is that we did not perform viral sequencing to determine the SARS-CoV-2 strains that infected our patients. Judging by the evolution of the COVID-19 pandemic, there are concerns over different viral strains acting differently and evading the immune system in such a way that more virulent variants, such as Delta, might affect pregnant women, their newborns, and vertical transmission in different ways. Since most of this study took place in 2020, we assume that the wild-type SARS-CoV-2 was involved in most cases [36], instead of the Delta variant that overcame in the second half of 2021. Moreover, we did not receive data on the viral load measured in each of the mediums and samples that were evaluated. Instead, the laboratory results returned only positive or negative answers.

Further study is needed to identify the long-term effects of SARS-CoV-2 infection during pregnancy. The long-term effects of COVID-19 infection in babies acquired during or after delivery should also be explored further.

## 5. Conclusions

Our data promote the theory that SARS-CoV-2 positive pregnant women do not expose their babies to a supplementary risk of infection via maternal nursing and breast milk feeding. Based on these results, we do not recommend the separation of the newborns from their mothers at birth since they do not seem to be at a higher risk of infection than the average.

## Figures and Tables

**Figure 1 jcm-10-05253-f001:**
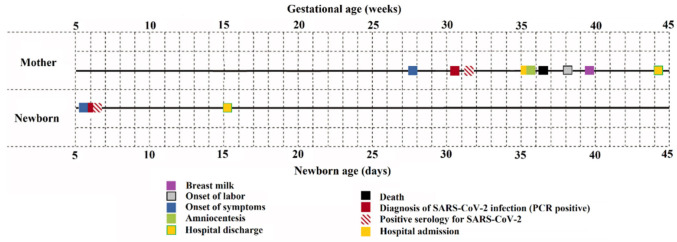
Timeline describing the average reference times for positive SARS-CoV-2 maternal and newborn PCR testing.

**Figure 2 jcm-10-05253-f002:**
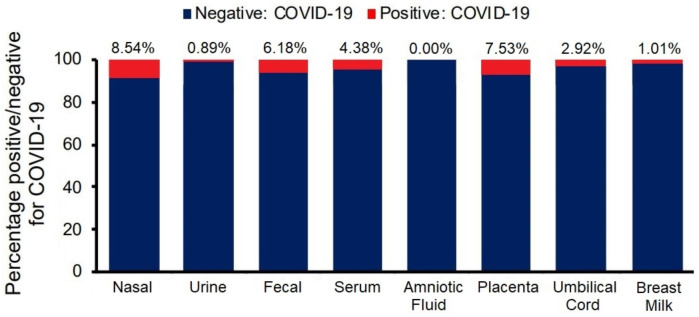
Maternal PCR test results.

**Figure 3 jcm-10-05253-f003:**
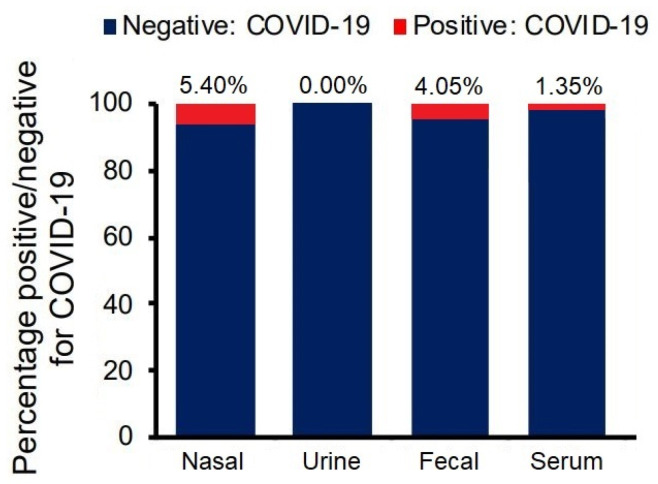
Neonatal PCR test results.

**Table 1 jcm-10-05253-t001:** General characteristics and outcomes of pregnant women and their neonates participating in this research as compared with a cohort of pregnant women without COVID-19.

Variables *	COVID-19 (*n* = 76)	Control (*n* = 813)	*p*-Value
**General characteristics**			
Age (years), mean (SD)	32.6 (4.6)	33.4 (5.2)	0.195
Pre-pregnancy BMI, mean (SD)	20.7 (2.6)	21.3 (3.6)	0.156
Pregnancies, mean (SD)	1.7 (0.6)	1.8 (0.6)	0.165
Gestations, mean (SD)	2.2 (0.6)	2.3 (0.7)	0.228
**Comorbidities ****			
Diabetes Mellitus	5 (6.5%)	58 (7.1%)	0.856
Asthma	4 (5.2%)	51 (6.3%)	0.726
Coagulation disorders	4 (5.2%)	42 (5.1%)	0.970
Arterial Hypertension	9 (11.8%)	94 (11.6%)	0.941
Thyroid disorders	5 (6.5%)	66 (8.1%)	0.635
UTI	7 (9.2%)	63 (7.7%)	0.651
**Maternal Outcomes**			
Pregnancy-associated DM	6 (7.9%)	28 (3.4%)	0.053
Gestational hypertension	7 (9.2%)	33 (4.1%)	0.038
Preeclampsia	4 (5.2%)	31 (3.8%)	0.534
Abnormal placental implantation	8 (10.5%)	64 (7.8%)	0.417
PROM	7 (9.2%)	53 (6.3%)	0.371
Emergency cesarean	18 (23.6%)	121 (14.9%)	0.043
ICU admission	4 (5.2%)	19 (2.3%)	0.124
Mortality	4 (5.2%)	14 (1.7%)	0.036
**Neonatal Outcomes (*n* = 74)**			
Prematurity	14 (18.9%)	72 (8.8%)	0.006
SGA	7 (9.2%)	32 (3.9%)	0.031
Malformations	6 (8.1%)	53 (6.5%)	0.644
Sepsis	4 (5.4%)	44 (5.4%)	0.952
ICU admission	3 (4.0%)	28 (3.4%)	0.819
Mortality	3 (4.0%)	21 (2.5%)	0.482
SARS-CoV-2 infection	4 (5.4%)	-	

* Data are presented as n(%) unless specified differently; ** Assessed before pregnancy; UTI—Urinary Tract Infections; DM—Diabetes Mellitus; SGA—Small for Gestational Age.

## Data Availability

The data presented in this study are available on request from the corresponding author.

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
