# Peer review of "Assessing SARS-CoV-2 Vertical Transmission and Neonatal Complications"

_jcm, 2021, doi:10.3390/jcm10225253_

Round 1

Reviewer 1 Report

Minor corrections

Introduction

- line 49, Conversely, the “hyperchromatic supranuclear stria” (SNS), corresponding to the Golgi apparatus, constitutes a marker for the anatomical and functional integrity of the ciliated cells; its rarefaction or disappearance during viral infections is a sign of cellular distressplease cite doi:10.1159/000508768

  • line 50, Covid-19 infection has an insidious course, often behaving in children or adolescents in a paucisymptomatic or asymptomatic manner, increasing the risk of contagion in the general population. please cite doi:10.12659/AJCR.925813
  • line 60, Evidence on several aspects of the prenatal management of these pregnancies remains mixed, including the type and frequency of fetal monitoring, the potential risk associated with invasive prenatal diagnosis, the timing of delivery, and intrapartum monitoring. please cite doi:10.1002/uog.23628
  • line 74, The placenta physiologically limits vertical transmission during pregnancy and has developed robust microbial defense mechanisms. It remains to this day whether the SARS-COV among the various microorganisms that cause congenital diseases may have evolved different mechanisms to circumvent these defenses.and please cite doi:10.1038/s41579-021-00610-y

Methods

  • line 148, please add references on vertical assessment
  • line 153, also here please add evidences from the literature
  • please add a figure describing the protocol and improve the description in the text, especially for selection criteria.

Results

well described and clear

Discussion

  • Symptoms of pregnant women with COVID-19 pneumonia were different, with the main symptoms being fever and cough. We found no evidence of vertical transmission in late pregnancy. The potential for vertical transmission of COVID-19 infection in pregnant women is relevant for understanding the clinical features and the significance and possible implications of the pathogen. please cite doi:10.1016/S0140-6736(20)30360-3
  • line 298, At present, there is limited information on the potential transmission of the infection from mother to child, particularly through breast milk and breastfeeding. A recent systematic review included 340 records, 37 with breast milk samples and 303 without. The 37 articles with breast milk samples analyzed reported 77 mothers nursing their infants and among them, 19 of 77 infants were confirmed COVID-19 cases based on RT-PCR tests, including 14 infants and five older children. Nine of 68 breast milk samples tested from mothers with COVID-19 were positive for SARS-CoV-2 RNA; of the exposed infants, four were positive and two were negative for COVID-19. Currently, there is no evidence of SARS-CoV-2 transmission through breast milk. The authors concluded that studies with longer follow-up periods that collect data on infant feeding practices and viral presence in breast milk are needed. please cite doi:10.1111/nyas.14477
  • line 313, The SARS-CoV2 pandemic has put a strain on healthcare systems around the world. The high volume of patients, combined with an increased need for intensive care and potential transmission, has forced the reorganization of hospitals and care delivery models. Standard operating procedures have been adapted for both facilities and healthcare professionals, including the development of well-defined and segregated patient care areas for the treatment of those affected by COVID-19. The availability of personal protective equipment (PPE) and adequate training of healthcare professionals in their use should be ensured as exposure to saliva suspensions, droplets and aerosols is increased in routine examination of the upper aero-digestive tract. please cite doi:10.23750/abm.v92i1.11281

Author Response

Dear reviewer,

Thank you for considering our manuscript, as we appreciate your efforts to analyse and improve our paper. Thus, based on the feedback received during the review phase, our team had carefully revised the article with the following changes:

  1. Thank you for recommending useful supplementary literature to be cited in text and strengthen our data. Therefore, we have included all the following suggestions:
    1. Lines 51-54: doi:10.1159/000508768
    2. Lines 54-57: doi:10.12659/AJCR.925813
    3. Lines 68-71: doi:10.1002/uog.23628
    4. Lines 75-79: doi:10.1038/s41579-021-00610-y
    5. Line 161: doi:10.23750/abm.v92i1.11281
    6. Lines 286-290: doi:10.1016/S0140-6736(20)30360-3
    7. Lines 291-301: doi:10.1111/nyas.14477
  2. Line 160 (previous 148): We have cited the WHO for vertical transmission assessment.
  3. Line 165 (previous 153): There was evidence from the Romanian Health Ministry regarding guidelines for COVID-19 maternal-neonatal interaction.
  4. The inclusion criteria was updated at lines 132 and 157.
  5. We performed other slight corrections inside the text, without causing alterations or changes of meaning.
  6. All edits were highlighted with the “track changes” function.

Best regards,

The authors

Reviewer 2 Report

A topic that has been studied before. However it is necessary to add more studies in the problem of vertical transmission from mother to the neonate.

The authors have shown that breast feeding does not lead to exposion of SARS to the neonates. The number of women included in the study is fair enough. The language is ok.

Author Response

Dear reviewer,

Thank you for considering our manuscript, as we appreciate your efforts to analyse and improve our paper.

Our team had carefully revised the article with the following changes:

  1. The abstract section was lightly modified.
  2. Supplementary literature was cited in text to strengthen our data:
    • Lines 51-54: doi:10.1159/000508768
    • Lines 54-57: doi:10.12659/AJCR.925813
    • Lines 68-71: doi:10.1002/uog.23628
    • Lines 75-79: doi:10.1038/s41579-021-00610-y
    • Line 161: doi:10.23750/abm.v92i1.11281
    • Lines 286-290: doi:10.1016/S0140-6736(20)30360-3
    • Lines 291-301: doi:10.1111/nyas.14477

  1. The introduction part was edited.
  2. We have clarified the inclusion criteria in the “materials and methods section”
  3. Line 160 (previous 148): We have cited the WHO for vertical transmission assessment.
  4. Line 165 (previous 153): There was evidence from the Romanian Health Ministry regarding guidelines for COVID-19 maternal-neonatal interaction.
  5. The inclusion criteria was updated at lines 132 and 157.
  6. We performed other slight corrections inside the text, without causing alterations or changes of meaning.
  7. All edits were highlighted with the “track changes” function.

Best regards,

The authors